# Comparative Efficacy of Intra-Articular Injection, Physical Therapy, and Combined Treatments on Pain, Function, and Sarcopenia Indices in Knee Osteoarthritis: A Network Meta-Analysis of Randomized Controlled Trials

**DOI:** 10.3390/ijms24076078

**Published:** 2023-03-23

**Authors:** Chun-De Liao, Hung-Chou Chen, Mao-Hua Huang, Tsan-Hon Liou, Che-Li Lin, Shih-Wei Huang

**Affiliations:** 1International Ph.D. Program in Gerontology and Long-Term Care, College of Nursing, Taipei Medical University, Taipei 110301, Taiwan; 2Department of Physical Medicine and Rehabilitation, Shuang Ho Hospital, Taipei Medical University, New Taipei City 235041, Taiwan; 3Department of Physical Medicine and Rehabilitation, School of Medicine, College of Medicine, Taipei Medical University, Taipei 110301, Taiwan; 4Department of Biochemistry, University of Washington, Seattle, WA 98015, USA; 5Department of Orthopedic Surgery, Shuang Ho Hospital, Taipei Medical University, New Taipei City 23561, Taiwan; 6Department of Orthopedics, School of Medicine, College of Medicine, Taipei Medical University, Taipei 11031, Taiwan

**Keywords:** sarcopenia, osteoarthritis, injection, physical therapy, pain, mobility, meta-analysis

## Abstract

Knee osteoarthritis (KOA) is associated with a high risk of sarcopenia. Both intra-articular injections (IAIs) and physical therapy (PT) exert benefits in KOA. This network meta-analysis (NMA) study aimed to identify comparative efficacy among the combined treatments (IAI+PT) in patients with KOA. Seven electronic databases were systematically searched from inception until January 2023 for randomized controlled trials (RCTs) reporting the effects of IAI+PT vs. IAI or PT alone in patients with KOA. All RCTs which had treatment arms of IAI agents (autologous conditioned serum, botulinum neurotoxin type A, corticosteroids, dextrose prolotherapy (DxTP), hyaluronic acid, mesenchymal stem cells (MSC), ozone, platelet-rich plasma, plasma rich in growth factor, and stromal vascular fraction of adipose tissue) in combination with PT (exercise therapy, physical agent modalities (electrotherapy, shockwave therapy, thermal therapy), and physical activity training) were included in this NMA. A control arm receiving placebo IAI or usual care, without any other IAI or PT, was used as the reference group. The selected RCTs were analyzed through a frequentist method of NMA. The main outcomes included pain, global function (GF), and walking capability (WC). Meta-regression analyses were performed to explore potential moderators of the treatment efficacy. We included 80 RCTs (6934 patients) for analyses. Among the ten identified IAI+PT regimens, DxTP plus PT was the most optimal treatment for pain reduction (standard mean difference (SMD) = −2.54) and global function restoration (SMD = 2.28), whereas MSC plus PT was the most effective for enhancing WC recovery (SMD = 2.54). More severe KOA was associated with greater changes in pain (β = −2.52) and WC (β = 2.16) scores. Combined IAI+PT treatments afford more benefits than do their corresponding monotherapies in patients with KOA; however, treatment efficacy is moderated by disease severity.

## 1. Introduction

Knee osteoarthritis (KOA), a prevalent joint disease that develops from degenerated articular cartilage, has become a growing problem in elderly populations. Among the clinical presentations of KOA, pain is the most prevalent symptom which directly affects the physical function of a patient’s lower limbs. Additionally, pain is associated with muscle weakness which is a common contributor to limitations on physical mobility and disease progression, especially in relation to walking capability [1,2]. Because aspects of physical mobility, such as walking speed and chair rise, are relevant indicators of frailty and sarcopenia in older individuals [3,4,5], developing effective treatment regimens for relieving pain, regaining leg strength, and recovering walking capability in older individuals with KOA is vital for the prevention of frailty and sarcopenia [6].

The primary goals of clinical management for KOA are pain relief, cartilage regeneration, and function recovery [7]. Intra-articular injection (IAI), involving agents such as corticosteroids (CSs), can provide moderate pain relief and minor functional improvement, albeit with some limitations. Physical therapy (PT) involving electric modality agents and exercise training has been identified as the most promising intervention for reducing pain and improving mobility in the early stages of KOA [8], despite poor treatment outcomes in patients with moderate to severe KOA. In addition, a number of biological agents have been developed and employed for cartilage repair in patients with KOA; among these biologics, hyaluronic acid (HA), platelet-rich plasma (PRP), and mesenchymal stem cells (MSCs) have been endorsed by clinical trials [9,10]. Among the multidisciplinary approaches for treating KOA, a combined treatment regimen of an IAI agent plus PT (IAI+PT) may be considered as the optimal strategy leading to significant improvement in pain and function in all disease stages until surgical treatment is required.

Systematic review and network meta-analysis (NMA) studies have investigated the relative efficacy of the following IAI agents, analyzed in pairs: CSs [11,12,13,14,15,16], HA [12,14,15,16,17,18,19,20,21,22,23,24,25], PRP [11,13,14,15,16,17,18,19,23,24,25], plasma rich in growth factor (PRGF) [12,13], botulinum neurotoxin type A (BoNTA) [12,18,23], ozone (OZ) [12,16,20,21], dextrose prolotherapy (DxTP) [18,23], MSCs [12,14,24,25,26], stromal vascular fraction of adipose tissue (SVF) [12,25,26], and autologous conditioned serum (ACS) [12,23,25]. However, none of these IAI agents have been comprehensively compared with others in a single NMA study, and thus, the overall relative efficacy of each remains unclear [12,14,18,24]. In addition, few systematic reviews have identified the combined treatment effects of IAI+PT [15,22,23]. A treatment model incorporating an IAI agent into a PT intervention seems promising; however, whether IAI+PT treatment yields any extra benefits compared with the IAI or PT monotherapies remains unclear.

The objectives of this NMA were to identify (1) the relative effects of multiple IAI+PT regimens on pain, global function, and walking capability; (2) the optimal treatment option by ranking the efficacy of each IAI+PT regimen; and (3) any relevant moderators for treatment outcomes.

## 2. Results

### 2.1. Selection of Studies

Figure 1 presents a flow diagram of the selection process. Through the electronic and manual literature searches, we identified 1125 relevant articles. After removing duplicates, we examined the titles and abstracts of 441 articles to assess their eligibility; after a review of the title or abstract of each article, 136 were considered relevant for full-text assessment. The final sample consisted of 80 RCTs published between 2001 and 2022 [27,28,29,30,31,32,33,34,35,36,37,38,39,40,41,42,43,44,45,46,47,48,49,50,51,52,53,54,55,56,57,58,59,60,61,62,63,64,65,66,67,68,69,70,71,72,73,74,75,76,77,78,79,80,81,82,83,84,85,86,87,88,89,90,91,92,93,94,95,96,97,98,99,100,101,102,103,104,105,106]. Four of the included RCTs [88,89,93,94] shared two registered clinical trials.

### 2.2. Characteristics of Analyzed Patients

Table 1 summarizes the demographic data and study characteristics of the included RCTs, and the details of each trial are presented in Appendix A. A total of 6934 patients were recruited with a mean (range) age of 59.6 (42.0–77.9) years, mean body mass index of 29.8 (23.3–34.3) kg/m^2^, and mean disease duration of 71 (10–307) months. The average proportion of women was 68%, whereas that of men was 33%. A total of 50.7% of the patients had a radiographic diagnosis of KL grade ≥ 3, and approximately half (42 out of 80) of the analyzed RCTs enrolled Asian populations.

In the present NMA, 56 of the analyzed RCTs had a two-arm design, and the other 24 were multiarm studies with a total of 186 study arms. Among all the patients, 3587 (51.7%) received IAI+PT, 738 (10.6%) received IAI alone, and 2438 (35.2%) received PT alone. Regarding the follow-up duration for the measurement outcomes, 70 RCTs had an immediate or short-term follow-up duration (range: 1 to 20 weeks), 48 had a medium-term follow-up duration (range: 24 to 36 weeks), and 27 had a long-term follow-up duration (range: 12 to 36 months; Appendix A).

### 2.3. Injection Treatment and Physical Therapy Characteristics

Ten types of IAI with 17 treatment options were identified in this NMA (Table 1), namely CSs (14 RCTs), BoNTA (four RCTs), HA (33 RCTs), OZ (ten RCTs), DxTP (nine RCTs), and several autologous biotics, namely PRP (20 RCTs), PRGF (two RCTs), ACS (two RCTs), SVF (one RCT), and MSCs (six RCTs). Regarding the injection protocol (Appendix A), most of the analyzed trials (54 RCTs, 70 arms) performed two to five injections with an interval of 1–4 weeks between each injection, whereas 23 RCTs (29 arms) prescribed a single injection. Two RCTs had ≥10 IAIs, and another two employed long intervals between each injection (12–16 weeks).

A total of 58 RCTs employed PT as monotherapy (59 RCTs) or combined therapy (66 RCTs). The PT protocol included exercise training, physical agent modality, and physical activity (Appendix A).

### 2.4. Quality and Risk of Bias in Analyzed Studies

The individual PEDro scores are listed in Appendix A. Overall, the methodological quality assessment revealed that 50 out of the 80 (62.5%) analyzed RCTs were classified as having high methodological quality (low risk of bias), whereas the other 30 were classified as having medium quality (unclear risk of bias), with a median PEDro score of 7/10 (range: 5/10 to 10/10). The interrater reliability of the cumulative PEDro scores was acceptable, with an intraclass correlation coefficient of 0.85 (95% CI: 0.77–0.91).

In total, 54 of the 80 (67.5%) analyzed RCTs employed a computer-based random assignment among which 36 RCTs had concealed allocation; in addition, 49 (61.3%) conducted an intention-to-treat analysis, and 75 (93.8%) had a dropout rate lower than 15% with respect to following the main outcomes. Next, 53 of the 80 (66.3%) analyzed RCTs adopted a blind methodology (Appendix A). In overall, there were high risks in selection bias, performance bias, detection bias, and attrition bias across studies.

### 2.5. Effectiveness of Treatment for Pain Reduction

Direct comparison results revealed that CS+PT (SMD = −1.31; 95% CI: −2.14, −0.49), DxTP+PT (SMD = −2.14; 95% CI: −2.90, −1.39), and HA+PT (SMD = −0.53; 95% CI: −1.02, −0.05) were more efficacious than was PT alone for pain reduction (Appendix A). Compared with HA+PT, the combined treatments BoNTA+PT (SMD = −3.94; 95% CI: −5.38, −2.50) and DxTP+PT (SMD = −3.18; 95% CI: −5.14, −1.21) resulted in greater pain-related changes; similar results were observed for PRP+PT (SMD = −2.16; 95% CI: −3.58, −0.74) compared with CS+PT (Appendix A).

The NMA for pain score was based on 78 RCTs with 18 treatment options and 110 pairwise comparisons (Figure 2A). The combined regimens, namely BoNTA+PT, DxTP+PT, HA+PT, MSC+PT, OZ+PT, PRGF+PT, and PRP+PT, resulted in favorable outcomes for pain reduction, with significant SMDs of −0.96 to −2.54, compared to UC during an overall follow-up timeframe (Figure 3). The global heterogeneity was significant (τ^2^ = 0.89, I^2^ = 93.6%, *p* < 0.0001). The node-splitting results for NMA revealed no inconsistencies between the direct and indirect evidence; the same findings were observed through visual inspection of a forest plot (Appendix A). Certainty of the evidence ranged from low to moderate among IAI+PT treatments, and generally very low to low among monotherapies (Appendix A). The most common reason for downgrading the certainty of evidence was related to major concerns about within-study bias, imprecision, and a small number of studies.

After the pooling of all the treatment effects in the NMA, the composite DxTP+PT was ranked the most effective (SUCRA = 0.93) of all the treatment arms for pain reduction, followed by acupoint BoNTA+PT (SUCRA = 0.91) and then PRGF+PT during an overall follow-up timeframe (SUCRA = 0.83; Figure 3). The subgroup analysis for each follow-up interval indicated that the combination treatments BoNTA+PT (SMD = −1.63; SUCRA = 0.90), MSC+PT (SMD = −1.80; SUCRA = 0.84), PRP+PT (SMD = −2.06; SUCRA = 0.86), and DxTP+PT (SMD = −5.36; SUCRA = 0.98) were the optimal options for pain reduction during the immediate, short- term, medium-term, and long-term follow-up interval, respectively (Appendix A). Additionally, the combined regimens (i.e., IAI+PT) generally achieved superior rankings to IAI and PT monotherapy during each follow-up timeframe, irrespective of the IAI type or PT program.

### 2.6. Effectiveness of Treatment for Global Function

Direct comparisons of pairwise meta-analyses indicated that composites BoNTA+PT, DxTP+PT, HA+PT, MSC+PT, PRP+PT, and SVF+PT achieved favorable effects on global function recovery compared with PT alone by corresponded SMDs of 0.47–1.60 during an overall follow-up period (Appendix A). In addition, the composites PRP+PT (SMD = 2.00; 95% CI: 0.89, 3.10) and BoNTA+PT (SMD = 1.04; 95% CI: 0.01, 2.07) obtained favorable effects compared with CS+PT and HA+PT, respectively.

The NMA for global function was based on 78 RCTs with 19 treatment regimens and 110 pairwise comparisons (Figure 2B). The results revealed significant effects in favor of all the combined regimens but not ACS+PT, with corresponded SMDs of 0.94–2.28, compared to UC during an overall follow-up timeframe (Figure 4). In addition, the global heterogeneity of the NMA model for global function was significant (τ^2^ = 0.47, I^2^ = 88.4%, *p* < 0.0001), and the node-splitting results did not indicate any relevant inconsistencies between the direct and indirect evidence (Appendix A). Certainty of the evidence generally ranged from very low to low among all treatment options (Appendix A). The most common reason for downgrading the certainty of evidence related to major concerns about within-study bias, imprecise, small number of studies, and potential publication bias.

After the pooling of all the treatment effects in the NMA, the composite DxTP+PT was ranked the most effective (SUCRA = 0.85) of all the treatment options for function recovery, followed by SVF+PT (SUCRA = 0.84) and PRGF+PT (SUCRA = 0.83) during an overall follow-up timeframe (Figure 4). The subgroup analysis based on the follow-up timeframe indicated that the composite treatment DxTP+PT yielded the highest probability of being the optimal treatment for function restoration during the immediate (SMD = 2.32; SUCRA = 0.89) and long-term follow-up interval (SMD = 3.38; SUCRA = 0.93); the composites PRP+PT (SMD = 2.04; SUCRA = 0.84) and PRGF+PT (SMD = 2.16; SUCRA = 0.84) were optimal during the short- and medium-term follow-up intervals, respectively (Appendix A). Additionally, the combined regimens (i.e., IAI+PT) generally achieved superiority over IAI and PT monotherapy during all follow-up intervals, irrespective of the IAI type or PT program.

### 2.7. Effectiveness of Treatment for Walking Capability

Direct comparisons of pairwise meta-analyses revealed that HA+PT had favorable effects on walking capability compared with PT alone (SMD = 0.61; 95% CI: 0.01, 1.21) and UC (SMD = 1.81; 95% CI: 0.86, 2.76) during an overall follow-up time interval (Appendix A).

The NMA for walking capability was based on 19 RCTs with 13 treatment regimens and 30 pairwise comparisons (Figure 2C). The IAI+PT regimens, namely CS+PT, DxTP+PT, HA+PT, MSC+PT, OZ+PT, PRGF+PT, and PRP+PT, achieved favorable effects on increasing walking capability, with corresponded SMDs of 1.57–2.54, compared to UC during an overall follow-up timeframe (Figure 4B). The global heterogeneity was significant (τ^2^ = 0.38, I^2^ = 83.8%, *p* < 0.001). The node-splitting results indicated no inconsistencies between the direct and indirect evidence (Appendix A). Certainty of the evidence ranged from low to moderate among combined treatment regimens whereas that ranged from very low to low among monotherapies (Appendix A). The most common reason for downgrading the certainty of evidence related to major concerns about within-study bias and small number of studies.

After the pooling of all the treatment effects in the NMA, the combined regimen MSC+PT was ranked the most effective (SUCRA = 0.84) of all the treatment options for walking capability during follow-up, followed by CS+PT (SUCRA = 0.72) and PRGF+PT (SUCRA = 0.64; Figure 5). The subgroup analysis of follow-up timeframe indicated that of all the treatment options, the composite HA+PT achieved the highest rank for increasing walking capability during the immediate (SMD = 1.30; SUCRA = 0.73) and short-term (SMD = 2.10; SUCRA = 0.89) follow-up timeframes (Appendix A); in addition, MSC+PT was optimal during the medium-term (SMD = 2.35; SUCRA = 0.79) and long-term (SMD = 4.61; SUCRA = 0.99) follow-up timeframes. Moreover, the combined regimens (i.e., IAI+PT) generally achieved superiority in terms of probability of the effects over IAI and PT monotherapy during each follow-up interval, irrespective of the IAI type or PT program (Appendix A).

### 2.8. NMR Results for Potential Moderators of Treatment Effects

The NMR results revealed that the KL grade 3–4 proportion of study sample was significantly associated with the SMDs for pain (β = −2.52; 95% CrI: −23.16 to −0.38) and walking capability (β = 2.16; 95% CrI: 1.05–3.23; Appendix A). In addition, sex (β = −8.15; 95% CrI: −12.56 to −2.88) as well as treatment composition of PT (β = 4.84; 95% CrI: 2.17–7.23) were significantly associated with effects for walking capability. Furthermore, a significant association was observed between population and treatment efficacy on global function (β = 1.68; 95% CrI: 0.33–3.09; Appendix A).

### 2.9. Compliance and Adverse Effects

Overall, a sample attrition rate of 0–53.8% was reported during follow-up on the basis of the analyzed RCTs, of which 0% to 30.8% were eliminated because of treatment noncompliance (Appendix A). The NMA results revealed no difference in compliance across all the IAI regimens with respect to UC (Figure 6A).

No serious adverse events related to treatment were reported after IAI alone or its combined treatments in any of the analyzed RCTs. In total, 54 of the 80 analyzed RCTs reported mild to moderate side effects, of which the most common were treatment-induced knee pain, joint stiffness, and effusion of short duration (Appendix A). With respect to UC, no significant adverse effects were observed among all of the IAI regimens (Figure 6B).

### 2.10. Publication Bias

A visual inspection of the comparison-adjusted funnel plot for publication bias comprising all the analyzed RCTs for each main outcome revealed no substantial asymmetry (Appendix A). The Begg–Mazumdar test results for pain reduction and walking capability revealed no reporting bias in any of the RCTs included in the NMA, whereas that for global function indicated significant reporting bias (*p* = 0.02).

## 3. Discussion

### 3.1. Summary of Main Findings

The results of the present study demonstrate that (1) a combined IAI+PT treatment regimen yields additional benefits for patients with KOA compared with monotherapies, regardless of the IAI agent and PT protocol involved; (2) the composite DxTP+PT was ranked the most effective strategy for pain reduction and global function recovery, whereas MSC+PT was the most optimal option for walking capability restoration; and (3) composite IAI+PT regimens generally achieved superior treatment effects compared with IAI or PT monotherapies, corresponding with an overall certainty of evidence ranging from very low to moderate. Next, the NMR results revealed that (1) disease severity based on the sample proportion of KL grade ≥3 may affect intervention outcomes related to pain and walking capability and (2) population and sex may have influences in treatment efficacy, particularly for global function and walking capability respectively. In addition, the IAI+PT regimens exhibited high compliance, as did the monotherapy regimens, despite the occurrence of nonserious adverse effects of IAI regimens.

### 3.2. Comparisons of this NMA with Previous Studies

Systematic reviews and NMAs have investigated the relative effects of multiple IAI monotherapies, and results have indicated that PRP yields superior treatment effects to those of HA [15,17,19,24,107,108], OZ [12], and CSs [11,12]. In addition, SVF exhibits favorable effects on pain compared with MSCs [12,26], as does HA compared with OZ [12,20,21], CSs [12], and MSCs [12,14]. However, other IAI agents, such as DxTP and PRGF, have yet to be comprehensively compared with conventional IAI agents. In the present study, a total of 10 IAI agents (i.e., CSs, BoNTA, HA, DxTP, OZ, PRP, PRGF, ACS, SVF, and MSCs) were identified and compared in an NMA and the results indicate that DxTP monotherapy exhibited greater treatment effects on pain and global function compared with other IAI monotherapies, namely MSCs, PRP, HA, OZ, and CSs (Figure 3, Figure 4 and Figure 5). Such results are generally consistent with those of other systematic reviews, particularly those investigated IAI monotherapy. In addition to the previous results, we identified that IAI combined with PT regimens can yield extra benefits compared with IAI alone in patients with KOA.

### 3.3. Explorations and Possible Mechanisms of Treatment Effects

In the present study, the results of direct and indirect treatment comparisons in meta-analysis revealed that DxTP+PT, HA+PT, MSC+PT, and PRP+PT achieved greater effects on all main outcomes than did PT alone. Our current findings indicate that IAIs of the viscosupplementation and autologous biogenetics categories yielded additional benefits for patients with KOA who were undergoing PT. The possible mechanisms driving effectiveness of these IAI interventions for KOA can be explained as follows.

A hypertonic DxTP reduces pain via nociceptive fiber transmission and by opening of the potassium channels [109,110,111], which induces an inflammatory response by recruitment of cytokines and growth factor and facilitates the tissue healing process [110,111,112,113]; in addition, blocking calcium and sodium electrolyte influx of the nociception receptor alongside decreasing substance P release can relieve the pain of KOA [100]. By contrast with DxTP, an HA injection exhibits mechanical effects, namely shock absorption and joint lubrication [114]. In addition, HA stimulates the syntheses of glycosaminoglycan and proteoglycan and exerts a chondroprotective effect through CD44 binding alongside the inhibition of interleukin (IL)-1β and matrix metalloproteinase production. Furthermore, HA produces an anti-inflammatory effect by suppressing the level of inflammatory factors, namely IL-1β, IL-6, IL-8, prostaglandin E2, and tumor necrosis factor [115]. Similarly, an MSC injection exhibits anti-inflammatory and immune modulation effects for osteoarthritis [116], and inhibits enthesophyte formation, synovitis, and cartilage degeneration [117]. Finally, PRP is an autologous blood product with a high concentration of platelets that is produced through the centrifugation of whole blood. The potential mechanism of PRP for KOA is tissue repair caused by growth factors and inflammatory mediators that stimulate cellular anabolism and exert anti-inflammatory effects [118].

The disease process of KOA impairs the tissues and structures surrounding the knee joint (i.e., articular cartilage, subchondral bone, ligaments, the periarticular muscles, and the synovium) resulted in inflammations and physical irritations of knee joints [119,120]. Based on the physiological effects described above, such IAIs can effectively treat the clinical features of KOA, a degenerative disorder characterized by pain and functional disability.

With respect to the therapeutic mechanism of PT, it mainly involves muscular strengthening, exercise therapy, electric physical agents, and gait modification. The primary positive effect of PT is to strengthen the lower-limb muscles, which alleviates instability and abnormal stress in the knee joint [121,122]. In addition, gait training through PT can correct the gait pattern of patients with KOA by reducing their knee load and pain [123,124]. Therefore, incorporating electric physical agents and exercise training into rehabilitation programs appears to obtain promising effects on restoration of walking ability, which further supports our results indicating that a mixed-component PT is significantly associated with greater treatment efficacy on walking capability.

In the present study, combined IAI+PT regimens generally exhibited superiority over monotherapies in terms of treatment outcomes. These findings indicate that IAIs combined with PT had a synergic effect in patients with KOA.

### 3.4. Moderator of Relative Efficiency among Treatment Regimens

Treatment effects in response to IAI regimens are likely affected by the degree of cartilage degeneration. Numerus studies have conducted subgroup analyses stratified by KL grade to identify the efficacy of IAI agents in patients with KOA of varying severity, particularly PRP [53,84,90,125,126,127,128], HA [38,90,127,128,129], ACS [73], and MSCs [130]. These studies have obtained conclusive results indicating that patients who suffer minor cartilage loss or have a low KL grade generally respond well to such IAI agents compared with those that receive a placebo injection or comparative treatment, whereas those who experience more severe cartilage degeneration or have a high KL grade experience less improvement through treatment. In addition, a higher radiological grade at baseline was significantly associated with a higher pain score and a higher WOMAC score after IAIs with PRP and OZ, respectively [57]. Contrary to the previous results, the present NMR results indicate that a higher KL grade (≥3) was associated with greater changes in pain and walking capability scores, indicating that patients with moderate to severe joint and cartilage degeneration are likely to respond better than are those with more minor cartilage loss. However, our findings are consistent with those of other studies [11,131]. In a systematic review, McLarnon et al. reported that differences in relative effects on pain reduction between PRP and CSs become more evident with increasing KOA severity (KL grade ≥3; SMD = −1.32) than with a low KL grade ≤2 (SMD = −0.08) [11]. Sucuoglu et al. reported that patients with a KL grade of 3 or 4 achieved greater pain reduction in response to PRP than did patients with a lower KL grade [131]. Kon indicated that patients with a region of full cartilage loss at baseline experienced considerable changes in WOMAC pain scores in response to ACS, particularly those who had more favorable baseline WOMAC pain scores [73]. The inconsistency between our results and those in the literature with respect to the treatment effects of IAIs in patients with KOA of varying severity may be due to the inclusion of patients with a wider range of disease severity in this study. However, additional systematic reviews and NMA studies are warranted to determine the differences in treatment effects among patient with low and high KL grades.

Another finding of particular interest showed that higher proportion of female patients was associated with poorer walking capability after treatment. Our findings indicate that sex may play a role in mediating relative treatment efficacy, particular the walking capability. In patients with KOA, sex differences have been observed in treatment outcomes of invasive managements such as radiofrequency and acupuncture treatments [132,133], as well as noninvasive therapies such as exercise training [134]. In agreement with the previous researches, our findings confirmed a gender effect in regulating relative treatment efficacy among invasive and noninvasive therapies (i.e., IAI and PT, respectively), and its combined regimens despite of that the underlying mechanism is yet to be elucidated. Some reasons may explain our findings as follows. First, gender is associated with distinct clinical phenotypes of KOA [135] and influences the variability of gait kinetics and kinematics in patients with KOA [136]. Therefore, treatment effects on walking capability can be affected by sex distribution of study sample, one of relevant confounding factors at baseline in an RCT. Second, female elder adults with KOA are more likely to report physical difficulty and impaired function in knee compared to their male counterparts [136]. Accordingly, female sex tends to be a potential risk factor of poor treatment outcome of physical mobility in KOA population. Finally, previous studies have identified older female patients achieve minor adaptations to exercise-based PT, in terms of muscle mass and strength gains compared to their male peers [137,138]. Especially, the sex-specific muscle morphological and functional adaptations responding to PT are apparent in the lower extremities, which may have further contributions in physical performance. Under such a scenario, women may experience poorer walking recovery than do men; and therefore, the higher the female proportion, the poorer the walking capability appears to be observed after treatments.

### 3.5. Certainty of the Evidence of Treatment Options for each Main Outcome

We found indications that IAI+PT was effective treatment strategy for KOA. However, the certainty of the evidence of IAI+PT regiments were mostly graded as moderate for pain and walking capability, and low for global function. The rank of the credibility of evidence for pain and walking capability was downgraded mainly because the high risk of biases (i.e., selection bias, performance bias, detection bias, and attrition bias) and the imprecise (wide confidence interval and small number of studies) domains assessed according to the GRADE system (Appendix A). Due to the identified publication bias, the credibility of evidence of each IAI+PT regimen for global function was generally lower than that for other main outcomes. In addition, the certainty of evidence among IAI and PT monotherapies was mostly ranked as very low for all outcomes due to that three or more domains were judged as serious consideration. According to the SUCRA and GRADE ranked results, the combined treatment IAI+PT not only yielded a greater superiority but stronger evidence than did its monotherapies. In line with the OARSI guidelines for the non-surgical management of KOA [139], IAI+PT should be recommended as the main option to treat KOA rather than IAI or PT alone, especially for relieving pain and enhancing walking capability [140].

### 3.6. The Needs of Multimodal Approach for Management of KOA

The clinical presentation of KOA is multifactorial and can be characterized by phenotypes across multiple dimensions including articular construction, biochemical markers, psychological distress, and patient characteristics (e.g., sex and body weight) [141,142,143]. Therefore, the complexity of the modern concept of KOA has been recognized as impairments of the whole joint, and not simply of joint cartilage [144]. This supports the needs of multimodal approach (i.e., the combined use of two or more interventions) to treat KOA as a whole joint disease and to meet patient expectations [145]. Results in the present NMA may strengthen the recommendations of multimodal approach by combining pharmacological and nonpharmacological treatments, particularly IAI+PT, in KOA [7,140,144,145]. Importantly, the comparable compliance and safety among combined regimens and its corresponding monotherapies identified by NMA in this study further underline the suggestion that the optimal use of IAI is attainable in combination with other interventions such as PT [140].

### 3.7. Strengths and Limitations

The strengths of this NMA include (1) full comparison of the relative effects among multiple IAI monotherapies as well as IAI+PT combined regimens, particularly the identified ten agents of IAI, in older people with KOA; (2) comprehensive assessment of risk of bias and methodological quality using PEDro scale; (3) identification of relevant moderator of treatment efficacy using NMR; and (4) grading the certainty of evidence in accordance with the GRADE approach. However, these strengths need to be balanced against the highly global heterogeneity across comparative arms in each of the main outcome.

This NMA has some limitations. First, because of the differences in modality agents and exercise types in PT regimens, providing a definitive conclusion for the effect of each type of PT on main outcomes was difficult. In addition, the number of injections as well as the production methods of biogenetics (e.g., low or high molecular weight for HA, leukocyte-poor or -rich PRP, MSCs derived from adipose tissue or bone marrow) for each IAI were not specified or independently analyzed in the NMA model. Therefore, we could not finalize an overall ranking regarding the superiority of certain IAI regimens. Finally, 25 of the 80 analyzed RCTs had a study sample size of ≤20; thus, the studies among these that reported no significant treatment effects on main outcomes may have contributed a negative effect size to the overall result.

## 4. Materials and Methods

### 4.1. Study Design and Protocol Registration

The present study was conducted in accordance with the Preferred Reporting Items for Systematic Reviews and Meta-Analysis (PRISMA) Extension Statement for NMA [146]. The protocol of this systematic review was registered in the PROSPERO registry (registration number: CRD42022336304). Relevant articles were identified through comprehensive electronic searches of several online databases—namely PubMed, EMBASE, CINAHL, the Cochrane Library, the Physiotherapy Evidence Database (PEDro), the China Knowledge Resource Integrated Database, and Google Scholar—until January 2023. In addition, we manually examined relevant systematic reviews for possible references. No limitations were imposed with respect to the publication year or language. All retrieved studies from search results were imported into Covidence electronic workflow platform [147], an internet-based collaboration platform that streamlines the trial selection process in a systematic-review study [148].

### 4.2. Search Strategy and Study Selection

The following keyword was used for patients’ conditions: “knee osteoarthritis.” The following keywords were used for IAI treatments: platelet rich plasma OR hyaluronic acid OR corticosteroid OR autologous conditioned plasma OR bone marrow aspirate OR ozone OR mesenchymal stem cell OR dextrose prolotherapy OR botulinum toxin. The following keywords were used for PT: physiotherapy OR exercise training OR physical activity OR “hydrotherapy/aquatic therapy” OR neuromuscular training OR vibration training OR electrotherapy OR shockwave therapy OR thermal therapy OR ultrasound OR neuromuscular electrical stimulation. The search formulae and keywords used for each database are presented in Appendix A.

### 4.3. Study Selection Criteria

Trials were included in the analysis if they met the following criteria: (1) the study was a randomized controlled trial (RCT) designed as parallel or cross-over settings; (2) the study enrolled the participants who had symptomatic or radiographic primary KOA; and the participants were excluded if they had comorbidities, including rheumatic arthritis, neurological diseases (e.g., spinal stenosis, stroke), or substantial abnormalities in hematological functions; (3) the trial had treatment arms of any IAI monotherapy or IAI+PT combination therapy; (4) a control arm receiving placebo IAI or usual care (UC), without any other IAI or PT, was used as the reference group in the present study; (5) the IAI treatments used anti-inflammatory drugs, such as CSs, analgesics (e.g., BoNTA, HA, DxTP, OZ mixed or not mixed with oxygen), or platelet derivatives (e.g., PRP, PRGF, ACS, SVF, MSCs); and (6) the PT involved rehabilitation treatments, such as exercise therapy, physical agent modalities (e.g., electrotherapy, shockwave therapy, thermal therapy), or physical activity training. Two researchers (CDL and SWH) independently performed study selection with disagreements resolved by discussions or involvement of a third reviewer (CLL), if necessary.

### 4.4. Outcome Measures

The primary outcomes of interest were pain score, global function, and walking capability. Pain score was measured using a quantifiable scale, namely a visual analog scale alongside a pain subscale of the Western Ontario and McMaster Universities Arthritis Index (WOMAC) and Knee Injury and Osteoarthritis Outcome Score (KOOS) [149].

Global function was assessed using self-report questionnaires [149], including the WOMAC physical difficulty subscale, KOOS physical function subscale, International Knee Documentation Committee score, Lequesne algofunctional index, and Lysholm knee score.

Walking capability, an indicator of sarcopenia [3,4,150], was assessed using a walking task, such as one with a 10 m walk, a timed up-and-go test, or a 6 min walk. The secondary outcome was adverse effects, which were assessed in terms of the number of patients reporting adverse events.

### 4.5. Data Collection and Extraction

The following data were extracted from each study and presented in an evidence table (Appendix A): (1) characteristics of the study design and sample, namely population area, age, body mass index, sex, disease duration, and disease severity presented as a Kellgren and Lawrence (KL) grade; (2) characteristics of the IAI+PT protocol; (3) follow-up time points; and (4) main outcome measures. The follow-up time intervals for the subgroup analysis were defined as immediate (<3 months), short (≥3 months, <6 months), medium (≥6 months, <12 months), and long (≥12 months); when multiple time points were reported within the same timeframe, the longest period was selected for analysis (e.g., if the follow-up time points for a pain score were 12 and 24 months, the data from the 24-month period were used for the long-term results). Data extraction was conducted by one researcher (CDL) and validated by another researcher (SWH). Any disagreement between these two researchers was resolved by a third researcher (THL).

### 4.6. Assessment of Bias Risk and Methodological Quality of Analyzed Studies

The PEDro scale was employed to evaluate the methodological quality of the retrieved RCTs [151]. The PEDro scale comprises 11 items, namely (1) eligibility criteria; (2) random allocation, (3) concealed allocation, (4) similarity at baseline, (5) subject blinding, (6) therapist blinding, (7) assessor blinding, (8) >85% follow-up for at least one key outcome, (9) intention-to-treat analysis, (10) between-group statistical comparison for at least one key outcome, and (11) point and variability measures for at least one key outcome. In accordance with the guidelines of the 11-item PEDro scale, the methodological quality of each RCT was rated by two researchers (CDL and SWH). The final PEDro score (range: 0–10) for each trial was obtained through a summation of the ratings for items 2 to 11 (score for each item: satisfactory = 1, unsatisfactory = 0). Any disagreements were resolved by a third researcher (THL).

The following five bias domains corresponded with ten judgement items of the PEDro scale of the analyzed RCTs were assessed: selection bias (items 2 and 3), performance bias (items 5 and 6), detection bias (item 7), attrition bias (items 8 and 9), and reporting bias (items 4, 10, and 11). On the basis of the final PEDro score, the methodological quality of each trial was classified as high (range: 7–10), medium (range: 4–6), or low (range: 0–3) [152]. The trial obtaining a ranked quality of medium or low was considered having an overall high risk of bias [153].

### 4.7. Data Synthesis and Analysis

Because of variation in the measurement tools for treatment outcomes among the trials, standard mean differences (SMDs) alongside 95% confidence intervals (CIs) were calculated to explore the treatment effect sizes of all the outcome measures across the trials. The SMD is defined as a pooled estimate of the mean difference between the change scores of any two of the study arms. The change scores were directly extracted whenever the mean and standard deviation (SD) scores of the pre–post change values were available. If the SD of the change score was not reported for the outcome measure, it was estimated on the basis of the baseline- and posttest–measured SD in accordance with the Cochrane Handbook for Systematic Reviews of Interventions [154]. We followed Rosenthal’s recommendations by assuming a pre–post correlation coefficient of 0.7 for a conservative estimation [155].

Direct and indirect comparisons among treatment regimens were made by running a random-effects NMA model within a frequentist framework. Heterogeneity and global consistency were assessed using the I^2^ statistic alongside τ^2^ values to estimate variance across the studies. The consistency between direct and indirect comparisons was assessed using the node-splitting method [156]. Ranking probabilities of effect estimation among treatments per outcome were expressed using the surface under the cumulative ranking (SUCRA) score [157].

Network meta-regression (NMR) models were used to identify any relevant moderators influencing heterogeneity across the studies. The NMA model for each outcome was adjusted using an individual moderator as a covariate [158]. Potential moderators were identified on the basis of (1) participant characteristics, namely age, body mass index, sex (i.e., proportion of women in the sample), population area, disease onset duration, and disease severity in terms of the KL grade 3–4 proportion of study sample (i.e., proportion of patients with KL grade ≥ 3 in the sample); and (2) the study methodology, comprising intervention design (i.e., monotherapy of IAI or its combined treatment with PT), treatment composition of PT (i.e., physical agent modality alone, exercise alone, or mixed components), PEDro score, treatment duration, and follow-up duration. The NMR results reported as *β* with 95% credible interval (CrI).

Next, compliance and adverse effects—measured in terms of the occurrence of treatment-related withdrawal and adverse events, respectively—were expressed as odds ratios (ORs) alongside 95% CIs. Finally, publication bias was assessed using funnel plots and the Begg–Mazumdar rank correlation test [159].

All analyses were conducted using R statistical software (version 4.0.4, R Foundation for Statistical Computing, Vienna, Austria) [158,160]. A two-tailed *p* value of < 0.05 was considered statistically significant for all the statistical analyses.

### 4.8. Certainty of Evidence

The Grading of Recommendation Assessment, Development, and Evaluation (GRADE) approach was used to determine confidence in an overall treatment ranking per outcome from the NMA [161]. The evaluation of evidence certainty began as high-quality evidence, and by evaluating the within-study bias, inconsistency, imprecision, incoherence, and publication bias, the quality of the evidence could be rated down to moderate, low, and very low. The evaluation procedures were performed in pairs and independently (CDL, HCC, THL, CLL, SWH).

## 5. Conclusions

The present NMA identified the comparative efficacy of multiple IAI+PT regimens and its monotherapies for older patients with KOA by accounting for potential biases related to selection, performance, and detection. The composite IAI+PT was generally superior to its corresponding monotherapy. Such relative effects among treatment regimens appear to be affected by disease severity, particularly in relation to pain and walking capability outcomes. Additionally, the composite DxTP+PT as well as MSC+PT was determined to be the optimal treatment strategy for pain and mobility outcomes, irrespective of the intervention mode or follow-up timeframe. The findings of this NMA could help guide the clinicians in prescriptions of IAI agents and PT to ensure optimal treatment outcomes for KOA in older individuals.

## Figures and Tables

**Figure 1 ijms-24-06078-f001:**
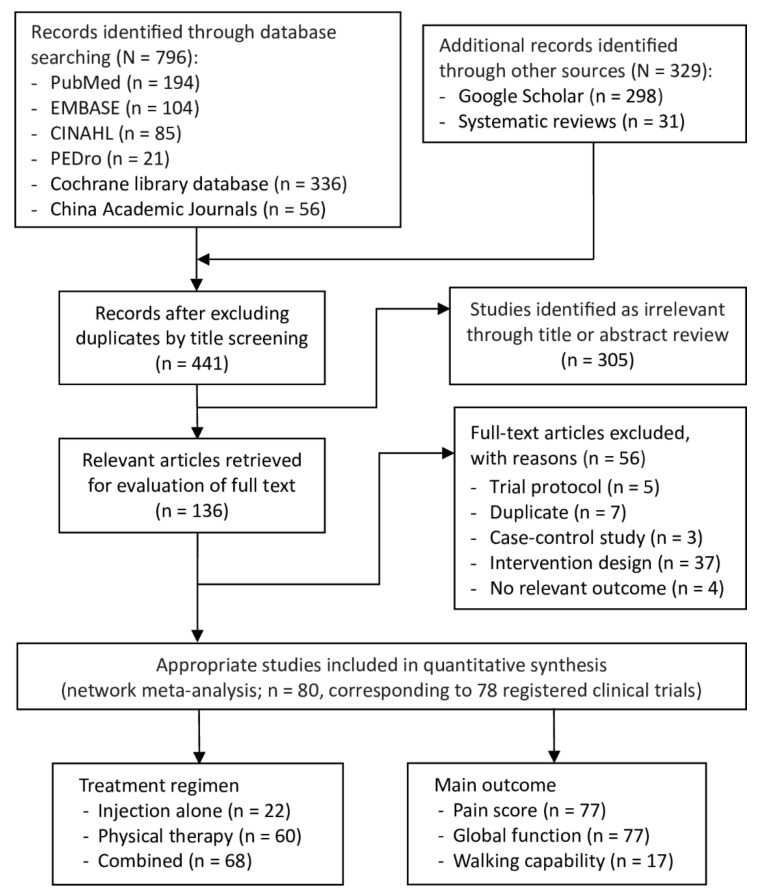
PRISMA flowchart of the study selection.

**Figure 2 ijms-24-06078-f002:**
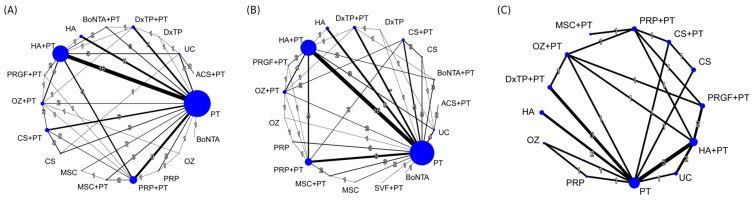
Network plot of direct comparisons among treatments for (**A**) pain, (**B**) global function, and (**C**) walking capability. The lines between nodes indicate direct comparisons in other studies. The size of each node is proportional to the number of participants. The thickness of each line is proportional to the number of studies denoted on the line. ACS, autologous conditioned serum; BoNTA, botulinum toxin type A; CS, corticosteroid; DxTP, dextrose prolotherapy; HA, hyaluronic acid; MSC, mesenchymal stem cell; OZ, ozone; PRP, platelet-rich plasma; PRGF, plasma rich in growth factor; PT, physical therapy; SVF, stromal vascular fraction; UC, usual care.

**Figure 3 ijms-24-06078-f003:**
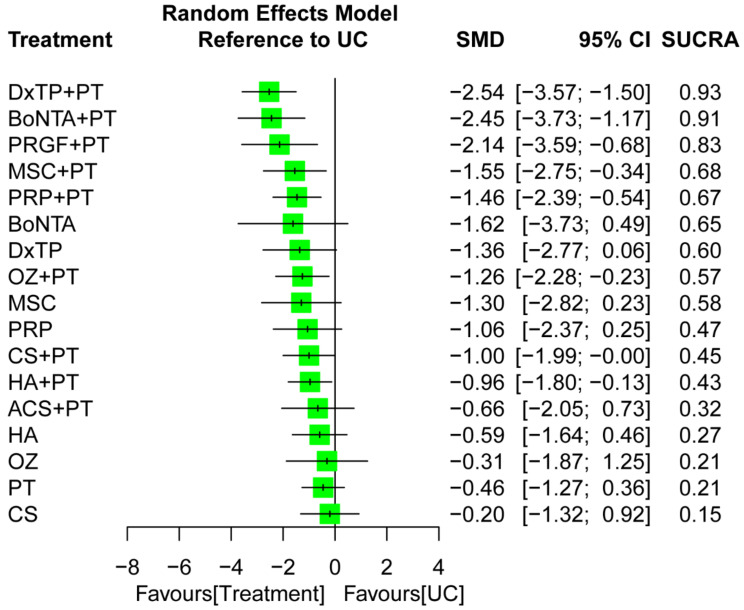
Forest plot summarizing the effects of treatment regimens on pain reduction for the entire follow-up duration. SMD, standardized mean difference; CI, confidence interval; SUCRA, surface under the cumulative ranking curve; ACS, autologous conditioned serum; BoNTA, botulinum toxin type A; CS, corticosteroid; DxTP, dextrose prolotherapy; HA, hyaluronic acid; MSC, mesenchymal stem cell; OZ, ozone; PRP, platelet-rich plasma; PRGF, plasma rich in growth factor; PT, physical therapy; UC, usual care.

**Figure 4 ijms-24-06078-f004:**
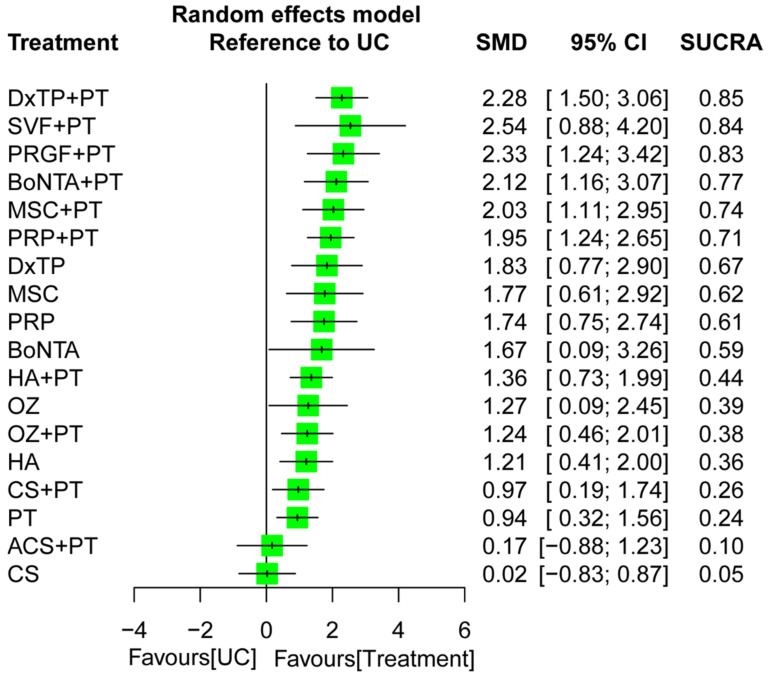
Forest plot summarizing the effects of treatment regimens on global function restoration for the entire follow-up duration. SMD, standardized mean difference; CI, confidence interval; SUCRA, surface under the cumulative ranking curve; ACS, autologous conditioned serum; BoNTA, botulinum toxin type A; CS, corticosteroid; DxTP, dextrose prolotherapy; HA, hyaluronic acid; MSC, mesenchymal stem cell; OZ, ozone; PRP, platelet-rich plasma; PRGF, plasma rich in growth factor; PT, physical therapy; SVF, stromal vascular fraction; UC, usual care.

**Figure 5 ijms-24-06078-f005:**
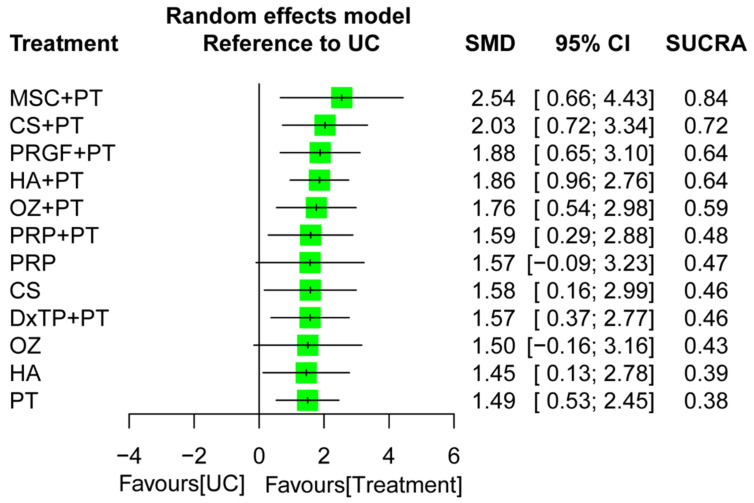
Forest plot summarizing the effects of treatment regimens on walking capability recovery for the entire follow-up duration. SMD, standardized mean difference; CI, confidence interval; SUCRA, surface under the cumulative ranking curve; CS, corticosteroid; DxTP, dextrose prolotherapy; HA, hyaluronic acid; MSC, mesenchymal stem cell; OZ, ozone; PRP, platelet-rich plasma; PRGF, plasma rich in growth factor; PT, physical therapy; UC, usual care.

**Figure 6 ijms-24-06078-f006:**
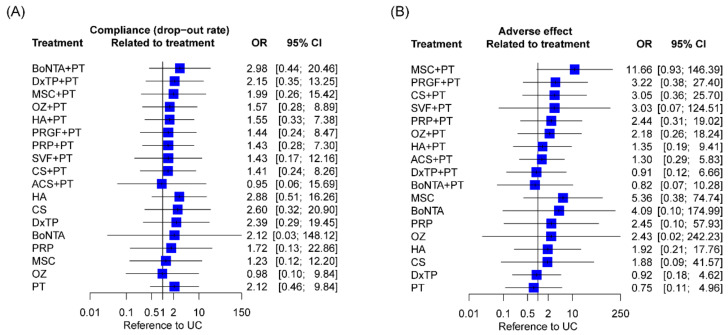
Compliance and adverse events of intra-articular infection regimens. Data concerning treatment-related (**A**) withdrawals and (**B**) adverse events were pooled using inverse variance weighting methods. OR, odds ratio; CI, confidence interval; ACS, autologous conditioned serum; BoNTA, botulinum toxin type A; CS, corticosteroid; DxTP, dextrose prolotherapy; HA, hyaluronic acid; MSC, mesenchymal stem cell; OZ, ozone; PRP, platelet-rich plasma; PRGF, plasma rich in growth factor; PT, physical therapy; SVF, stromal vascular fraction; UC, usual care.

**Table 1 ijms-24-06078-t001:** Study summary.

	Injection Alone			Physical Therapy Alone			Combined Treatment			Usual Care		
	Trials (Groups), *n* ^a^	Sample (*n*)	Mean (Range) ^a^	Trials (Groups), *n* ^a^	Sample (*n*)	Mean (Range)^a^	Trials (Groups), *n* ^a^	Sample (*n*)	Mean (Range) ^a^	Trials (Groups), *n* ^a^	Sample (*n*)	Mean (Range) ^a^
All included trials	23 (26)	748		59 (62)	2438		65 (91)	3577		7 (7)	171	
Age, year ^b^	22 (25)	738	60.3 (51–78)	58 (61)	2417	60.5 (42–75)	65 (91)	3566	59.4 (42–76)	7 (7)	171	57.9 (49–65)
BMI, kg/m^2 b^	10 (14)	490	28.1 (24.5–32.6)	40 (42)	1847	29.9 (24.4–34.3)	48 (70)	2864	29.2 (23.7–34.3)	1 (1)	28	32.7
Gender, n (%)												
Male	17 (19)	180	28% (10–55%)	44 (46)	644	32% (3–81%)	52 (80)	1166	36% (3–87%)	7 (7)	70	38% (7–70%)
Female	18 (20)	468	73% (45–100%)	46 (48)	1402	70% (19–100%)	53 (81)	2123	65% (13–100%)	7 (7)	101	62% (40–93%)
Population area, n												
America	3 (3)	134		15 (15)	880		15 (19)	995		2 (2)	57	
Europe	8 (9)	184		6 (6)	234		14 (18)	714		2 (2)	44	
Asia	11 (12)	399		32 (35)	1191		32 (47)	1708		2 (2)	50	
Africa	2 (3)	60		4 (4)	94		2 (3)	119		1 (1)	20	
Oceania	0			1 (1)	10		2 (4)	41		0		
Disease duration, month ^b^	11 (13)	420	56 (12–85)	26 (28)	1158	76 (13–306)	26 (39)	1659	70 (10–307)	2 (2)	64	91 (5–144)
KL grade, n (%)												
≤2	14 (15)	294	51.3% (0–100%)	46 (48)	949	47.5% (0–100%)	61 (73)	1590	50.3% (0–100%)	4 (4)	51	69.2% (48–100%)
≥3	14 (15)	239	45.5% (0–100%)	46 (48)	1150	49.9% (0–100%)	53 (73)	1698	52.3% (0–100%)	4 (4)	63	55.7% (29–100%)
Intervention regimen, n (compliance, %) ^b^												
Injection therapy	22 (25)	738		0			66 (92)	3587		0		
CS	5 (5)	138					12 (12)	377				
BoNTA	1 (1)	25					3 (4)	137				
HA	9 (11)	316					25 (25)	1462				
OZ	2 (2)	44					9 (9)	332				
DxTP	2 (2)	62					8 (9)	292				
Autologous biotics ^c^	5 (5)	163					26 (32)	977				
Physical therapy	0			59 (62)	2438		66 (92)	3587		0		
Exercise				28 (30)	813		39 (60)	1890				
Physical agent modality				14 (14)	425		7 (7)	241				
Physical activity				3 (3)	54		7 (10)	315				
Clinical characteristics (baseline) ^b^												
Pain status												
VAS (0–100)	14 (16)	461	65.8 (28.0–85.2)	44 (44)	1757	61.4 (30.0–93.5)	46 (64)	2464	66.8 (32.9–97.1)	6 (6)	142	54.3 (33.0–83.8)
WOMAC–pain (0–20)	9 (10)	293	9.4 (4.8–13.9)	28 (29)	1036	9.8 (3.6–17.3)	35 (47)	1873	9.0 (3.0–18.9)	4 (4)	92	11.1 (7.5–13.3)
Global function												
WOMAC–PF (0–68)	8 (9)	261	45.7 (4.8–13.9)	30 (31)	1067	39.0 (12.3–70.7)	36 (49)	1823	36.0 (17.7–79.9)	4 (4)	92	37.9 (33.4–45.3)
KOOS–PF (0–100)	3 (3)	75	40.7 (34.4–48.3)	10 (10)	370	52.8 (34.7–77.0)	13 (19)	669	56.8 (33.7–75.4)	2 (2)	43	46.3 (44.0–48.6)
Walking speed, m/s	1 (2)	40	0.54 (0.51–0.56)	9 (10)	330	0.87 (0.59–1.37)	13 (18)	455	0.93 (0.69–1.89)	0		

^a^ Number of trials that reported the indicated item. ^b^ All summations calculated on the basis of the values reported in the analyzed studies that could be estimated. ^c^ Treatment regimens included platelet-rich plasma, mesenchymal stem cells, plasma rich in growth factor, autologous conditioned serum, and stromal vascular fraction. BMI, body mass index; KL grade, Kellgren and Lawrence grading system for classification of osteoarthritis; CS, corticosteroid; BoNTA, botulinum toxin type A; HA, hyaluronic acid: OZ, ozone; DxTP, dextrose prolotherapy; VAS, visual analog scale; WOMAC, Western Ontario and McMaster Universities Osteoarthritis scale; WOMAC–PF, Western Ontario and McMaster Universities Osteoarthritis–physical function; KOOS-PF, Knee injury and Osteoarthritis Outcome Score–physical function.

## Data Availability

Refer to Appendix A. Raw data available on request.

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
