# Peer review of "Comparative Efficacy of Intra-Articular Injection, Physical Therapy, and Combined Treatments on Pain, Function, and Sarcopenia Indices in Knee Osteoarthritis: A Network Meta-Analysis of Randomized Controlled Trials"

_ijms, 2023, doi:10.3390/ijms24076078_

Round 1

Reviewer 1 Report

Overall, I think this was a very well-done meta-analysis on an interesting topic. It is clear that much effort went into the production of this manuscript, and I thank the authors for their preparation. Following the PRISMA guidelines and registering with PROSPERO were significant strengths of the methodology. 

I have some minor comments related to how the authors report their findings and conclusions, in addition to some grammatical notes. 

Section 4.1: What software was used to screen the articles for inclusion/exclusion into the meta-analysis (Covidence, Rayyan, etc.)?

Section 4.2: The text of the paragraph overlaps with that of the line numbering. 

Line 316: Change "have not yet to be" to "have yet to be"

Line 336 - Change "block" to "blocking" ?

Lines 356-363: The manuscript is light on the physical therapy content, and I am wondering whether there is a need to go back to the evidence on the main categories of PT included in the study to discuss how this contributes to patient improvement. Much more effort is made to demonstrate the effectiveness of the IAI than the PT component. 

Line 376: Change "contralaterally" to "Contrary"

Line 398: period needed after KOA

Lines 444-446 need to be edited, as I think the author repeats themselves twice on the same statement, once without including author initials.

Comments related to discussion: The authors point out in their results that KL grade and sex moderate the results; however, only KL grade is discussed specifically. I think the effect of sex should be considered as well. I think the authors also need to spend more time in the discussion on the effects of bias (PEDro scores) and certainty (GRADE) on the overall interpretation of the findings. Essentially discuss the implications of Table S4 to clinical interpretation. 

Author Response

Comments and Suggestions for Authors

Overall, I think this was a very well-done meta-analysis on an interesting topic. It is clear that much effort went into the production of this manuscript, and I thank the authors for their preparation. Following the PRISMA guidelines and registering with PROSPERO were significant strengths of the methodology.

I have some minor comments related to how the authors report their findings and conclusions, in addition to some grammatical notes.

Response

We thank all the reviewers for their comprehensive review and their comments regarding our manuscript. We have made all necessary modifications to our originally submitted manuscript (Manuscript ID: ijms-2274985), based on reviewers’ comments, point by point.

Section 2.1: What software was used to screen the articles for inclusion/exclusion into the meta-analysis (Covidence, Rayyan, etc.)?

Response

Thank you for the constructive comment. We used Covidence software to perform the study selection process. The statements were made in the revised manuscript as follows:

Section 2.1 (Lines 99–101).

“All retrieved studies from search results were imported into Covidence systematic review software (Veritas Health Innovation, Melbourne, VIC, Australia), an internet-based collaboration platform that streamlines the study selection process.”

Section 2.2: The text of the paragraph overlaps with that of the line numbering.

Response

The indicated format is corrected accordingly.

Line 316: Change "have not yet to be" to "have yet to be"

Response

We revised the statement as follows:

Section 4.2 (Lines 463–464).

“However, other IAI agents, such as DxTP and PRGF, have yet to be comprehensively compared with conventional IAI agents.”

Line 336 - Change "block" to "blocking" ?

Response

The statement is revised as follows:

Section 4.3 (Lines 484–485).

“in addition, blocking calcium and sodium electrolyte influx of the nociception receptor alongside decreasing substance P release can relieve the pain of KOA [48].”

Lines 356-363: The manuscript is light on the physical therapy content, and I am wondering whether there is a need to go back to the evidence on the main categories of PT included in the study to discuss how this contributes to patient improvement. Much more effort is made to demonstrate the effectiveness of the IAI than the PT component.

Response

Because varieties (ten types) of IAI agents were identified, we served PT as a whole to simplify the types of combined treatment (i.e., IAI+PT) in NMA. According to the reviewer’s insightful comments, we consider the main categories of PT as one of moderators. Accordingly, we performed additional meta-regression analyses to identify associations of PT category with relative treatment efficacy for each outcome. Thereby, we obtained additional findings and made related statements in the Methods, Results, and Discussion sections, which are summarized as follows:

Materials and Methods

Section 2.7 (Lines 206–209).

“(2) the study methodology, comprising intervention design (i.e., monotherapy of IAI or its combined treatment with PT), treatment composition of PT (i.e., physical agent modality alone, exercise alone, or mixed components), ...”

Results

Section 3.8 (Lines 408–410).

“In addition, sex (β = −8.15; 95% CrI: −12.56 to −2.88) as well as treatment composition of PT (β = 4.84; 95% CrI: 2.17‒7.23) were significantly associated with effects for walking capability.”

Discussion

Section 4.3 (Lines 510–513).

“Therefore, incorporating electric physical agents and exercise training into rehabilitation programs appears to obtain promising effects on restoration of walking ability, which further supports our results indicating that a mixed-component PT is significantly associated with greater treatment efficacy on walking capability.”

Line 376: Change "contralaterally" to "Contrary"

Response

The word is revised as follows:

Section 4.4 (Line 529).

“Contrary to the previous results, ...”

Line 398: period needed after KOA

Response

The statement is revised as follows:

Section 4.7 (Lines 606–607).

“particularly the identified 10 agents of IAI, in older people with KOA; (2) …”

Lines 444-446 need to be edited, as I think the author repeats themselves twice on the same statement, once without including author initials.

Response

The statement is revised as follows:

Section 2.3 (Lines 127–129).

“Two researchers (CDL and SWH) independently performed study selection with disagreements resolved by discussions or involvement of a third reviewer (CLL), if necessary.”

Comments related to discussion: The authors point out in their results that KL grade and sex moderate the results; however, only KL grade is discussed specifically. I think the effect of sex should be considered as well. I think the authors also need to spend more time in the discussion on the effects of bias (PEDro scores) and certainty (GRADE) on the overall interpretation of the findings. Essentially discuss the implications of Table S4 to clinical interpretation.

Response

Following the reviewer’s constructive comment, we made statements to elucidate sex difference in treatment effect as follows:

Section 4.4 (Lines 546–569).

“Another finding of particular interest showed that higher proportion of female patients was associated with poorer walking capability after treatment. Our findings indicate that sex may play a role in mediating relative treatment efficacy, particular the walking capability. In patients with KOA, sex differences have been observed in treatment outcomes of invasive managements such as radiofrequency and acupuncture treatments [73, 74], as well as noninvasive therapies such as exercise training [75]. In agreement with the previous researches, our findings confirmed a gender effect in regulating relative treatment efficacy among invasive and noninvasive therapies (i.e., IAI and PT, respectively), and its combined regimens despite of that the underlying mechanism is yet to be elucidated. Some reasons may explain our findings as follows. First, gender is associated with distinct clinical phenotypes of KOA [76] and influences the variability of gait kinetics and kinematics in patients with KOA [77]. Therefore, treatment effects on walking capability can be affected by sex distribution of study sample, one of relevant confounding factors at baseline in an RCT. Second, female elder adults with KOA are more likely to report physical difficulty and impaired function in knee compared to their male counterparts [77]. Accordingly, female sex tends to be a potential risk factor of poor treatment outcome of physical mobility in KOA population. Finally, previous studies have identified older female patients achieve minor adaptations to exercise-based PT, in terms of muscle mass and strength gains compared to their male peers [78, 79]. Especially, the sex-specific muscle morphological and functional adaptations responding to PT are apparent in the lower extremities, which may have further contributions in physical performance. Under such scenario, women may experience poorer walking recovery than do men; and therefore, the higher the female proportion, the poorer the walking capability appears to be observed after treatments.”

According to the reviewer’s comment, we also made an additional subsection in the discussion section to explore the effects of bias and certainty (GRADE) on the overall interpretation of the findings. The implications of Table S5 (original Table S4) to clinical interpretation are also mentioned as follows:

Section 4.5 (Lines 572–587).

“We found indications that IAI+PT was effective treatment strategy for KOA. However, the certainty of the evidence of IAI+PT regiments were mostly graded as moderate for pain and walking capability, and low for global function. Rank of the credibility of evidence for pain and walking capability was downgraded mainly because the high risk of biases (i.e., selection bias, performance bias, detection bias, and attrition bias) and the imprecise (wide confidence interval and small number of studies) domains assessed according to the GRADE system (Supplementary Table S5). Due to the identified publication bias, the credibility of evidence of each IAI+PT regimen for global function was generally lower than that for other main outcomes. In addition, the certainty of evidence among IAI and PT monotherapies was mostly ranked as very low for all outcomes due to that three or more domains were judged as serious consideration. According to the SUCRA and GRADE ranked results, the combined treatment IAI+PT not only yielded a greater superiority but a stronger evidence than did its monotherapies. In line with the OARSI guidelines for the non-surgical management of KOA [80], IAI+PT should be recommended as the main option to treat KOA rather than IAI or PT alone, especially for relieving pain and enhancing walking capability [81].”

Reviewer 2 Report

Dear authors,

I have read your article entitled "Comparative Efficacy of Intra-Articular Injection, Physical Therapy, and Combined Treatments on Pain and Sarcopenia Indices in Knee Osteoarthritis" with great interest. I found your study to be well-organized and informative, and I believe that your findings make a valuable contribution to the field of osteoarthritis treatment. Below are my comments on your article:

1. The title does not mention the focus on sarcopenia indices in KOA, which is an important aspect of the study. Optional - a more accurate title could be "Comparative Efficacy of Intra-Articular Injection, Physical Therapy, and Combined Treatments on Pain, Function, and Sarcopenia Indices in Knee Osteoarthritis. Besides, PRISMA guidelines recommend mentioning the type of review – systematic review and meta-network analysis: “Identify the report as a systematic review, meta-analysis, or both”.

2. The abstract seems insufficient in information and volume. The abstract lacks important information on the inclusion and exclusion criteria used for selecting the RCTs and on the types of intra-articular injections analyzed. This could be important for readers to assess the validity and generalizability of the study findings from the abstract.

3. The introduction of your article clearly identifies the research gap and highlights the importance of your study. It provides a good overview of the current state of knowledge in the field and sets the context for your research question.

4. The methodology of your study is sound and robust, and the use of network meta-analysis is appropriate for this type of research. You have also taken care to report the methods and results in a transparent and clear manner.

5. Your results provide useful insights into the comparative efficacy of different treatment options for KOA. The finding that combined IAI + PT treatments offer more benefits than their corresponding monotherapies is particularly noteworthy.

6. The use of meta-regression analyses to explore potential moderators of treatment efficacy is a valuable addition to your study. The finding that disease severity moderates treatment efficacy provides important information for clinicians and researchers.

7. The discussion section of your article is well-written and clearly explains the implications of your findings. You have also highlighted the limitations of your study and provided suggestions for future research.

8. Relevant articles should be discussed presenting OA as a whole joint/body disease. After all, our understanding of osteoarthritis (OA) has evolved to recognize its complex interaction with the human body as a whole, necessitating a redefinition of the disease concept. Thus, any future conservative disease-modifying treatment for knee osteoarthritis (KOA) should adopt a multimodal, holistic approach that considers the osteoarthritis joint in the context of the entire body, and not limit itself only to intraarticular injections [e.g.: PMID: 32803403].

Overall, I think that your study is well-conducted and provides valuable insights into the comparative efficacy of different treatment options for KOA. I appreciate your contributions to this field and look forward to seeing future studies building on your work.

Author Response

Comments and Suggestions for Authors

Dear authors,

I have read your article entitled "Comparative Efficacy of Intra-Articular Injection, Physical Therapy, and Combined Treatments on Pain and Sarcopenia Indices in Knee Osteoarthritis" with great interest. I found your study to be well-organized and informative, and I believe that your findings make a valuable contribution to the field of osteoarthritis treatment. Below are my comments on your article:

Response

We thank all the reviewers for their comprehensive review and their comments regarding our manuscript. We have made all necessary modifications to our originally submitted manuscript (Manuscript ID: ijms-2274985), based on reviewers’ comments, point by point.

  1. The title does not mention the focus on sarcopenia indices in KOA, which is an important aspect of the study. Optional - a more accurate title could be "Comparative Efficacy of Intra-Articular Injection, Physical Therapy, and Combined Treatments on Pain, Function, and Sarcopenia Indices in Knee Osteoarthritis. Besides, PRISMA guidelines recommend mentioning the type of review – systematic review and meta-network analysis: “Identify the report as a systematic review, meta-analysis, or both”.

Response

Thank you for the constructive comment. We revised the title as follows:

“Comparative Efficacy of Intra-Articular Injection, Physical Therapy, and Combined Treatments on Pain, Function, and Sarcopenia Indices in Knee Osteoarthritis: A Network Meta-Analysis of Randomized Controlled Trials”

  1. The abstract seems insufficient in information and volume. The abstract lacks important information on the inclusion and exclusion criteria used for selecting the RCTs and on the types of intra-articular injections analyzed. This could be important for readers to assess the validity and generalizability of the study findings from the abstract.

Response

We added information regarding selection criteria of RCTs in the abstract as follows:

“All RCTs which had treatment arms of IAI agents [autologous conditioned serum, botulinum neurotoxin type A, corticosteroids, dextrose prolotherapy (DxTP), hyaluronic acid, mesenchymal stem cells (MSC), ozone, platelet-rich plasma, plasma rich in growth factor, and stromal vascular fraction of adipose tissue] in combination with PT [exercise therapy, physical agent modalities (electrotherapy, shockwave therapy, thermal therapy), and physical activity training] were included in this NMA. A control arm receiving placebo IAI or usual care, without any other IAI or PT, was used as the reference group.”

  1. The introduction of your article clearly identifies the research gap and highlights the importance of your study. It provides a good overview of the current state of knowledge in the field and sets the context for your research question.
  2. The methodology of your study is sound and robust, and the use of network meta-analysis is appropriate for this type of research. You have also taken care to report the methods and results in a transparent and clear manner.
  3. Your results provide useful insights into the comparative efficacy of different treatment options for KOA. The finding that combined IAI + PT treatments offer more benefits than their corresponding monotherapies is particularly noteworthy.
  4. The use of meta-regression analyses to explore potential moderators of treatment efficacy is a valuable addition to your study. The finding that disease severity moderates treatment efficacy provides important information for clinicians and researchers.
  5. The discussion section of your article is well-written and clearly explains the implications of your findings. You have also highlighted the limitations of your study and provided suggestions for future research.

  1. 8. Relevant articles should be discussed presenting OA as a whole joint/body disease. After all, our understanding of osteoarthritis (OA) has evolved to recognize its complex interaction with the human body as a whole, necessitating a redefinition of the disease concept. Thus, any future conservative disease-modifying treatment for knee osteoarthritis (KOA) should adopt a multimodal, holistic approach that considers the osteoarthritis joint in the context of the entire body, and not limit itself only to intraarticular injections [e.g.: PMID: 32803403].

Response

Following the reviewer’s insightful comment, we made an additional subsection in the discussion section to elucidate the multimodal, holistic approach for KOA as follows:

Section 4.6 (Lines 590–602).

“The clinical presentation of KOA is multifactorial and can be characterized by phenotypes across multiple dimensions including articular construction, biochemical markers, psychological distress, and patient characteristics (e.q., sex and body weight) [82-84]. Therefore, the complexity of the modern concept of KOA has been recognized as impairments of the whole joint, and not simply of joint cartilage [85]. This supports the needs of multimodal approach (i.e., the combined use of of two or more interventions) to treat KOA as a whole joint disease and to meet patient expectations [86]. Results in the present NMA may strengthen the recommendations of multimodal approach by combining pharmacological and nonpharmacological treatments, particularly IAI+PT, in KOA [7, 81, 85, 86]. Importantly, the comparable compliance and safety among combined regimens and its corresponding monotherapies identified by NMA in this study further underline the suggestion that the optimal use of IAI is attainable in combination with other interventions such as PT [81].”
